# Portable and accurate diagnostics for COVID-19: Combined use of the miniPCR thermocycler and a well-plate reader for SARS-CoV-2 virus detection

**Everardo González-González**[1,2], **Grissel Trujillo-de Santiago**[1,3], **Itzel Montserrat Lara-Mayorga**[1,2], **Sergio Omar Martínez-Chapa**[3], **Mario Moisés Alvarez**[1,2] *

**1** Centro de Biotecnología-FEMSA, Escuela de Ingeniería y Ciencias, Tecnologico de Monterrey, Monterrey, Nuevo León, México, **2** Departamento de Bioingeniería, Escuela de Ingeniería y Ciencias, Tecnologico de Monterrey, Monterrey, Nuevo León, México, **3** Departamento de Ingeniería Mecátrónica y Eléctrica, Escuela de Ingeniería y Ciencias, Tecnologico de Monterrey, Monterrey, Nuevo León, México

\* mario.alvarez@tec.mx

**Data Availability Statement:** All relevant data are within the manuscript and at MedRxiv: https://

## Abstract

The coronavirus disease 2019 (COVID-19) pandemic has crudely demonstrated the need for massive and rapid diagnostics. By the first week of July, more than 10,000,000 positive cases of COVID-19 have been reported worldwide, although this number could be greatly underestimated. In the case of an epidemic emergency, the first line of response should be based on commercially available and validated resources. Here, we demonstrate the use of the miniPCR, a commercial compact and portable PCR device recently available on the market, in combination with a commercial well-plate reader as a diagnostic system for detecting genetic material of the severe acute respiratory syndrome coronavirus 2 (SARS-CoV-2), the causal agent of COVID-19. We used the miniPCR to detect and amplify SARS-CoV-2 DNA sequences using the sets of initiators recommended by the World Health Organization (WHO) for targeting three different regions that encode for the N protein. Prior to amplification, samples were combined with a DNA intercalating reagent (i.e., EvaGreen Dye). Sample fluorescence after amplification was then read using a commercial 96-well plate reader. This straightforward method allows the detection and amplification of SARS-CoV-2 nucleic acids in the range of ~625 to $2 \times 10^5$ DNA copies. The accuracy and simplicity of this diagnostics strategy may provide a cost-efficient and reliable alternative for COVID-19 pandemic testing, particularly in underdeveloped regions where RT-QPCR instrument availability may be limited. The portability, ease of use, and reproducibility of the miniPCR makes it a reliable alternative for deployment in point-of-care SARS-CoV-2 detection efforts during pandemics.

## Introduction

Recent epidemic events (i.e., Zika in Southeast Asia and Latin-America in 2016 [1,2], Ebola in West Africa in 2013–2015 [3], and pandemic Influenza A/H1N1/2009 [4]) have clearly

www.medrxiv.org/content/10.1101/2020.04.03.
20052860v1.supplementary-material.

**Funding:** The authors acknowledge the funding
provided by the Federico Baur Endowed Chair in
Nanotechnology (0020240I03). EGG acknowledges
funding from a doctoral scholarship provided by
CONACyT (Consejo Nacional de Ciencia y
Tecnología, México). GTdS, EGG, and MMA
acknowledge the institutional funding received
from Tecnológico de Monterrey (Grant
002EICIS01). MMA, GTdS, SOMC and IMLM
acknowledge funding provided by CONACyT
(Consejo Nacional de Ciencia y Tecnología, México;
https://www.conacyt.gob.mx/) through grants SNI
26048, SNI 256730, SNI 31803, SNI 228923,
respectively. The funders had no role in study
design, data collection and analysis, decision to
publish, or preparation of the manuscript.

**Competing interests:** The authors have declared
that no competing interests exist.

evidenced the urgent need for low-cost, portable, and easy-to-use diagnostic systems that can be effectively deployable to address epidemic episodes [5–8]. However, these portable diagnostic systems have been mainly viewed as solutions for underprivileged or remote places and/or for catastrophic scenarios. Nevertheless, the coronavirus disease 2019 (COVID-19) pandemic caused by the severe acute respiratory syndrome coronavirus 2 (SARS-CoV-2) [9] has broadsided most well developed and developing countries with only a few (i.e., South Korea [10], China, Singapore [11], and Taiwan [12]) showing an ability to deploy massive efforts for rapid and accurate detection of positive infection cases. The swift and massive testing of thousands of possibly infected subjects has been an important component of the strategy of these countries that has helped to effectively mitigate the spreading of COVID-19 among their populations [10,12–14]. And yet, most nations are still struggling to implement massive testing [15–17]. Current testing methods have exhibited important limitations in widespread reach, flexibility, cost-effectiveness, and scalability during this pandemic.

Through the last two pandemic events involving influenza A/H1N1/2009 and COVID-19 [18], the Centers for Disease Control (CDC) and the World Health Organization (WHO) recommended the reverse transcription quantitative polymerase chain reaction (RT-qPCR) as the gold standard for official detection of positive cases. Without any doubt, nucleic acid amplification, and particularly RT-qPCR, is the most reliable technique for the early and accurate detection of viral diseases [19,20]. Unfortunately, conducting RT-qPCR diagnostics often depends on access to centralized laboratory facilities for testing [21–23]. To resolve this limitation, several different versions of compact PCR platforms (some of them q-PCR systems) have been described recently in the scientific literature [24–27]. Unfortunately, most of these devices have not yet reached the market. During epidemic emergencies, resourcing of incompletely developed technologies is impractical, and the use of commercially available diagnostic platforms becomes the first and arguably the most cost-efficient line of defense.

Only recently, several miniaturized PCR machines become commercially available [28,29]. One of them, the miniPCR [30,31], reached the international market in 2015. The most recent version of this compact PCR machine has an approximate cost of ~$800 USD (www.minipcr.com) as compared to $3000 USD for a conventional PCR thermocycler [28]. Several papers have documented the value of the miniPCR® system as a portable and robust diagnostic tool [32–36]. We recently published a comparison of the performance of the miniPCR and a commercial thermal cycler for the identification of artificial Zika and Ebola genetic sequences. Our experiments using a wide variety of primers sets and template concentrations revealed no differences in performance between either thermal cycler type [37]. The commercial availability, low price (as compared to conventional thermocyclers), portability, and user friendliness of the miniPCR makes it an attractive and tangible solution that effectively brings PCR analysis to the POC. In the present study, we demonstrate the convenience of using the miniPCR for the detection and amplification of synthetic samples of SARS-CoV-2 [18], the causal viral agent of the current COVID-19 pandemic.

## Materials and methods

### Equipment specifications

We ran several sets of amplification experiments in a miniPCR from Amplyus (MA, USA). The unit has dimensions of $20 \times 5 \times 15$ cm, weighs 0.7 kg, and requires 120V (AC) and 3.5 A to operate. The miniPCR can run 8–16 amplifications in parallel (depending on the model employed).

A commercial power supply (PowerPac from Bio-Rad, CA, USA) was used to operate the electrophoresis unit used to run the agarose gels to reveal the amplification products obtained

by the miniPCR thermocycler. A Bio-Rad ChemiDoc XRS imaging system was used for end-point PCR detection. Alternatively, the miniPCR unit has its own blueGel electrophoresis unit (Fig 1A and 1B), a compact electrophoresis unit (23 × 10 × 7 cm) that weighs 350 g, that is powered by a built-in power supply (AC 100–240 V, 50–60 Hz).

We also used a Synergy HT microplate reader (BioTek Instruments, VT, USA) to detect the fluorescence induced by an intercalating reagent in positive samples from the PCR reactions.

## Controls for validation

We used a plasmid containing the complete N gene from 2019-nCoV, SARS, and MERS as positive controls at a concentration of 200,000 copies/μL (Integrated DNA Technologies, IA, USA). Samples containing different concentrations of synthetic nucleic acids of SARS-CoV-2 were prepared by successive dilutions from stocks containing 200,000 copies $mL^{-1}$ ng/L of viral nucleic acids. We used a plasmid containing the GP gene from Ebola Virus (EBOV) as a negative control. The production of this EBOV genetic material has been documented previously by our group [37].

## Amplification mix

We used REDTaq Ready Mix from Sigma-Aldrich (USA), and followed the recommended protocol: 10 μL Readymix, 0.5 μM of forward primer, 0.5 μM of reverse primer,1μL of DNA template (~ 625 to 2x105 DNA copies), 1μl of EvaGreen Dye, and nuclease free water to final volume of reaction 20 μL.

**Primers used.** Three different sets of primers were used to target three different regions of the SARS-CoV-2 N gene sequence. These primer sets are identical to those recommended by the Center of Disease Control (CDC) for the standard diagnostics of COVID-19 (i.e., N1, N2, and N3 assays) using quantitative real time PCR. Sequences of all these primers and their corresponding amplicons are presented in Tables 1 and 2.

## Amplification protocols

For all PCR experiments, we used the same three-stage protocol (see Fig 1D) consisting of a denaturation stage at 94˚C for 5 min, followed by 25 cycles of 94˚C for 20s, 60˚C for 30s, and 72˚C for 20s, and then a final stage at 72˚C for 5 min, for a total duration of 60 minutes in the miniPCR® thermocycler.

## Documentation of PCR products

We analyzed 10 μL of each PCR product using 2% agarose electrophoresis in Tris-acetic acid-EDTA (TAE) buffer (Sigma-Aldrich, MO, USA). Gels were dyed with GelGreen (Biotium, CA, USA) using a 1:10,000 dilution and a current of 110 V supplied by a Bio-Rad PowerPac HC power supply (Bio-Rad, CA, USA) for 40 min. We used the Quick-Load Purple 2-Log DNA ladder (NEB, MA, USA) as a molecular weight marker. We analyzed the gels by UV transillumination using a Bio-Rad ChemiDoc XRS imaging system. In some of our experiments, we also used the blueGel unit, a portable electrophoresis unit sold by MiniPCR from Amplyus (MA, USA). In these experiments, we analyzed 10 μL of PCR product using 2% agarose electrophoresis tris-borate-EDTA buffer (TBE). Gels were dyed with Gel- Green (CA, USA) using a 1:10,000 dilution, and a current of 48 V was supplied. Photo-documentation was done using a smartphone camera. As a third method of detection and to read the amplification product, we evaluated the amplification products by detecting the fluorescence emitted by a DNA intercalating agent, the EvaGreen® Dye, in the Synergy HT microplate reader (BioTek

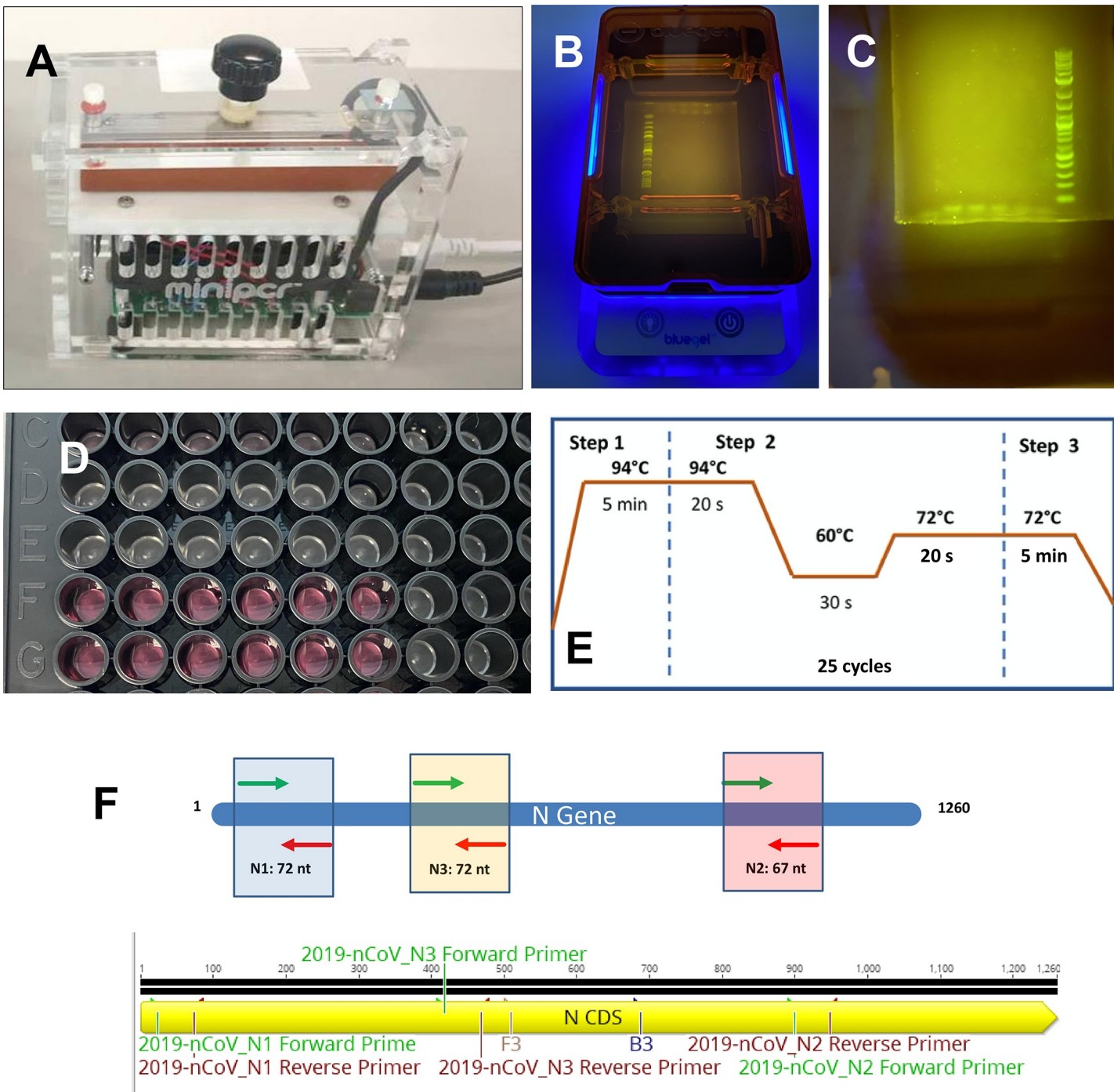

**Fig 1. Equipment and initiators for SARS-CoV-2 detection using a miniPCR.** A) The miniPCR® thermocycler. (B) The blueGel® electrophoresis chamber: blueGel® allows visualization of a 15 ml agarose gel using an integrated blue LED array. (C) Agarose gel electrophoresis of the SARS-CoV-2 amplification products. (D) Commercial 96-well plate with COVID-19 controls (artificial DNA samples). (E) Temperature cycling scheme used in our PCR protocol. (F) Three different sets of primers were used to target a gene sequence encoding the SARS-CoV-2 N protein.

Instruments, VT, USA). Briefly, 20 μL of the PCR reaction mix was placed in distinct wells of a 96-well plate, after completion of the PCR program. Each well was made to a final volume of 150 μL by adding 130 μL of nuclease free water and the samples were well mixed by pipetting.

**Table 1. Primer sequences used in PCR amplification experiments.**

| Name | Description | Primers Sequence (5'>3') |
|---|---|---|
| *2019-nCoV_N1-F* | 2019-nCoV_N1 Forward Primer | 5'-GAC CCC AAA ATC AGC GAA AT-3' |
| *2019-nCoV_N1-R* | 2019-nCoV_N1 Reverse Primer | 5'-TCT GGT TAC TGC CAG TTG AAT CTG-3' |
| *2019-nCoV_N2-F* | 2019-nCoV_N2 Forward Primer | 5'-TTA CAA ACA TTG GCC GCA AA-3' |
| *2019-nCoV_N2-R* | 2019-nCoV_N2 Reverse Primer | 5'-GCG CGA CAT TCC GAA GAA-3' |
| *2019-nCoV_N3-F* | 2019-nCoV_N3 Forward Primer | 5'-GGG AGC CTT GAA TAC ACC AAA A-3' |
| *2019-nCoV_N3-R* | 2019-nCoV_N3 Reverse Primer | 5'-TGT AGC ACG ATT GCA TTG-3' |

These experiments were run in triplicate. The following conditions were used in the microplate reader: excitation of 485/20, emission of 528/20, gain of 75. Fluorescence readings were made from the above at room temperature.

## Statistical analysis

Determination of mean values and standard deviations were conducted using Excel tools. All experiments were run by triplicate. Regression analysis was conducted in Excel.

## Results and discussion

Time is the most limiting factor in epidemic emergencies. Therefore, the integration of well-developed and commercially available technologies [31,37,38] becomes an obvious, expedient, and cost-effective first line of defense in the context of COVID-19 pandemics. Here, we demonstrate that the combined use of a commercial and portable PCR unit (the miniPCR) and a 96-well plate reader is potentially adequate for the fast deployment of diagnostic efforts. We show the combined ability of both units to amplify and identify different synthetic genetic sequences of SARS-CoV-2 (see Materials and Methods).

### Analysis of sensitivity

We conducted a series of experiments to assess the sensitivity of the PCR reactions conducted in the miniPCR thermocycler using the three sets of primers recommended by CDC to diagnose infection by SARS-CoV-2. Table 1 shows the sets of primers used to target genetic sequences that code for the expression of the SARS-CoV-2 N protein. Table 2 shows the sequence of the DNA products (amplicons) generated by successful targeting of these regions with the N1, N2, and N3 primer pairs.

Fig 2A–2C show the PCR products of the amplification reactions conducted using three different primer pairs. In all cases, different concentrations of SARS-CoV-2 genetic material, in the range of $2.0 \times 10^5$ to 625 DNA copies, were used as reaction templates. If we put this range in the proper clinical context, the actual viral load of COVID-19 in nasal swabs from patients has been estimated to fall within the range of $10^5$ to $10^6$ viral copies per mL [18]. The amplification proceeds with sufficient quality to allow proper visualization of the amplification products in electrophoresis gels, even at low nucleic acid concentrations. Fig 2A–2C shows agarose gels

**Table 2. Amplicon sequences generated (and their corresponding lengths) by each of the primer pairs used in the PCR amplification experiments.**

| Primer pair | Amplicon sequence | Amplicon Length (nt) |
|---|---|---|
| *N1* | GACCCCAAAATCAGCGAAATGCACCCCGCATTACGTTTGGTGGACCCTCAGATTCAACTGGCAGTAACCAGA | 72 |
| *N2* | TTACAAACATTGGCCGCAAATTGCACAATTTGCCCCCAGCGCTTCAGCGTTCTTCGGAATGTCGCGC | 67 |
| *N3* | GGGAGCCTTGAATACACCAAAAGATCACATTGGCACCCGCAATCCTGCTAACAATGCTGCAATCGTGCTACA | 72 |

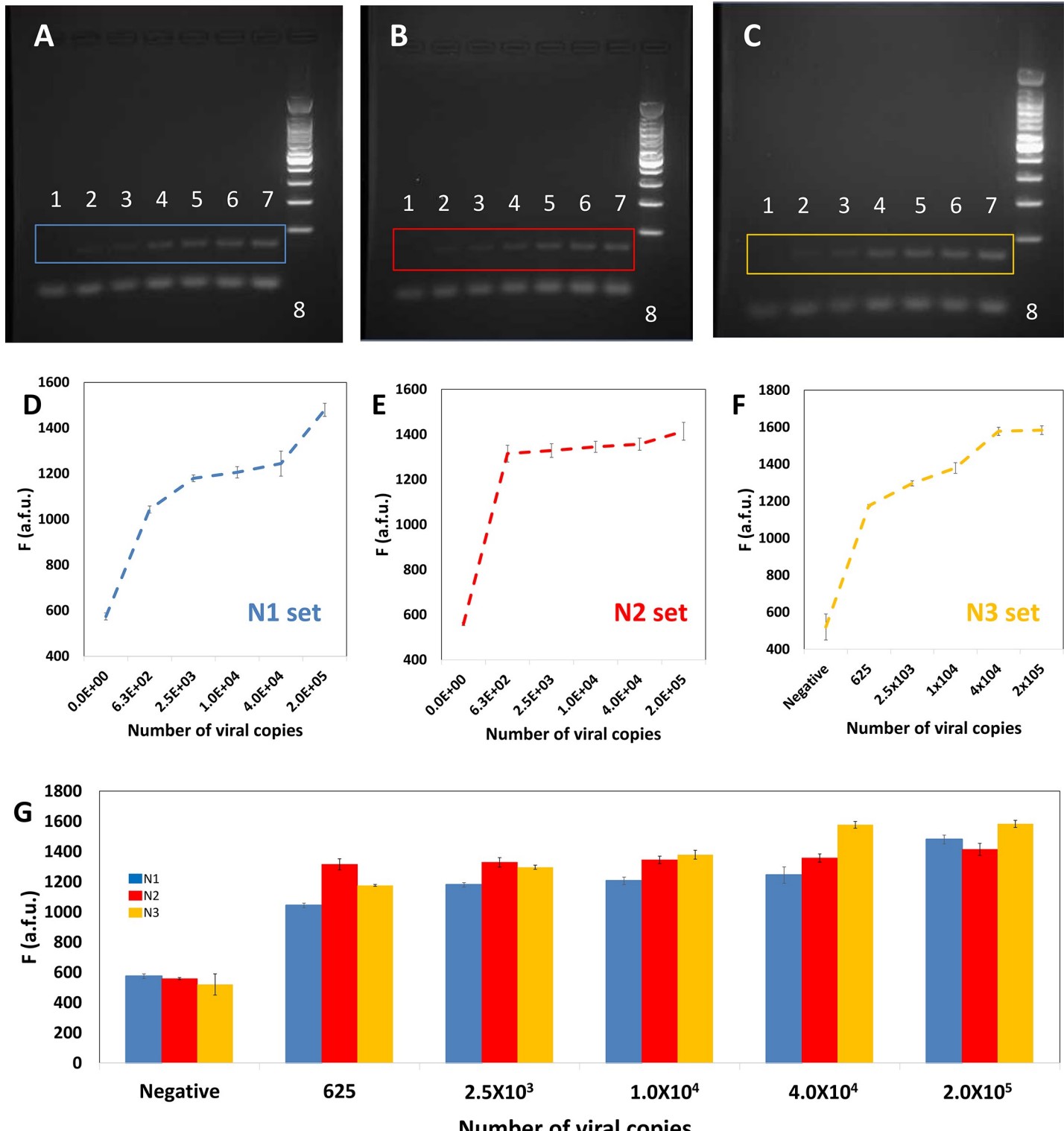

**Fig 2. Evaluation of the sensitivity of the combined use of a miniPCR® thermal cycler (for amplification) and a plate reader (for determination of the amplification extent).** (A-C) Sensitivity trials using different concentrations of the template (positive control) and three different primers sets (A) N1, indicated in blue; (B) N2, indicated in red; and (C) N3, indicated in yellow. Images of agarose gel electrophoresis of the DNA amplification product generated by targeting three different regions of the sequence coding for SARS-CoV-2 N protein. PCR was performed using a miniPCR® thermocycler. Three different primer sets were used (N1, N2, and N3). The initial template amount was gradually increased from left to right: negative control (lane 1), 625 copies (lane 2), $2.5 \times 10^3$ (lane 3), $1.0 \times 10^4$ (lane 4), repetition

of $1.0 \times 10^4$ (lane 5), $4.0 \times 10^4$ (lane 6), $2.0 \times 10^5$ DNA copies (lane 7), and molecular weight ladder (lane 8). (D-F) Determination of fluorescence, as measured in a commercial plate reader, for different dilutions of SARS-CoV-2 synthetic DNA templates. Results using three different primer sets are shown: (A) N1, indicated in blue; (B) N2, indicated in red; and (C) N3, indicated in yellow. (G) Summary and comparison of fluorescence readings form synthetic samples of SARS-CoV-2 in a wide span of dilutions. Results using three different primer sets are shown: (A) N1, indicated in blue; (B) N2, indicated in red; and (C) N3, indicated in yellow.

containing the amplification products of each one of three experiments, where the three different sets of primers (namely N1, N2, and N3) were used to amplify the same range of concentrations of template. The miniPCR® was able to generate a visible band of amplification products for all three primer sets and across the whole range of synthetic viral loads.

In general, the products of amplification in final point PCR are primarily detected on agarose gels using conventional electrophoresis techniques conducted with conventional lab equipment. The miniPCR® system is commercialized with its own electrophoretic unit ("blueGel®"; Fig 1B and 1C). The blueGel® has several important advantages and represents a valid and portable solution for detecting PCR amplification products. Nevertheless, running an experiment aimed at visualizing amplification products, as with any standard gel electrophoresis procedure requires time. A good separation of bands typically involves a processing time of 35 to 60 minutes from the loading of the amplification product to the final documentation through photography.

As an alternative, we show here that the amount of amplification product can be quantitatively evaluated using a commercial 96-well plate reader. To do this, we used an intercalating agent during amplification in the miniPCR apparatus. Fig 2D–2F shows the fluorescence readings associated with the analysis of the different dilutions of synthetic SARS-CoV-2 samples previously revealed by gel electrophoresis (S1 Fig, S2 Fig and S3 Fig). We ran triplicate reactions for each dilution and for each primer data set. The fluorescence readings were capable of clearly discriminating between positive and negative samples across the whole range of dilutions tested (from $2 \times 10^5$ to 625 copies). This observation holds true for each of the three primer sets tested. Note that the use of a plate reader, instead of a conventional gel electrophoresis unit, presupposes a significant savings in time. Up to 96 PCR reactions can be read in a matter of 5 to 10 minutes. This implies that an array of 12 miniPCR units and a plate reader could equal the throughput of a traditional RT-QPCR platform, but at one third of the capital cost. In addition, during emergencies and particularly in developing countries, attaining or buying regular thermal cyclers and plate readers is much easier than purchasing or accessing RT-qPCR systems.

In addition, our results suggest that fluorescence readings using a plate reader exhibit high reproducibility and robustness. Overall, we obtained small standard deviations (in the range of 6 to 40 arbitrary fluorescence units [a.f.u.]) and a small average variance coefficient (2.6%) in fluorescence readings across the whole range of values of viral copies tested. We observed similar variability indicators in experiments using different primer pairs. For instance, we observed variance coefficients of 2.31%, 2.15%, and 3.34% when using primer sets N1, N2, and N3, respectively. If we considered only fluorescence readings from positive samples, we observed variance coefficients of 2.23%, 2.34%, and 1.31% when using primer sets N1, N2, and N3, respectively.

Fig 2G consolidates the fluorescence readings obtained from miniPCR amplifications using synthetic SARS-CoV-2 samples and the primer sets N1 (blue bars), N2 (red bars), and N3 (yellow bars). Overall, this data set is consistent. These results suggest that any of the primer sets tested (N1, N2, or N3) may be used to amplify SARS-CoV-2 genetic material in the miniPCR. However, for the experimental conditions tested (i.e., the nature and concentration of the intercalating agent, the concentrations of primers, and the concentration of enzyme, among

others), we observe differences in the performance of each primer pair. For example, primer sets N1 and N3 appear to promote amplifications in which the observed fluorescence is proportional to the initial concentration of DNA template (i.e., the viral load). By contrast, primer pair N2 appears to generate amplification product with high fluorescence emissions even at low values of the initial final copy numbers. Note that all fluorescence readings for positive samples shown in Fig 2E exhibit a fluorescence reading between 1300 and 1400 a.f.u.

Furthermore, measuring the fluorescence with the plate reader may add a quantitative element to the analysis of positive COVID-19 samples. In principle, samples with higher viral loads will exhibit higher fluorescence if processed through the same PCR program (i.e., exposed to the same number of cycles). For example, for amplifications using primer set N3, we observe a linear relationship between the natural logarithm of the number of viral copies and the natural logarithm of fluorescence signal for the range of 625 to 40,000 viral copies:

$$\mathrm{Ln\ (viral\ load)} = \alpha * \mathrm{Ln(F_{sample} - F_o)} \qquad (1)$$

where $F_o$ is the fluorescence reading exhibited by a blank (i.e., a negative sample prepared and processed in the same way than the positive samples) and $\alpha = 8.897$ (as determined by fitting of the data presented in Fig 3A). For instance, we believe we can adjust the concentration of intercalating reagent to assure linearity of the fluorescence signal with respect to the viral load for experiments with different primer sets. This simple strategy will result in a fully quantitative, reliable, and easily implemented quantitative version of a straightforward final-point PCR protocol.

Using the primers and methods described here, we were able to consistently detect the presence of SARS-CoV-2 synthetic DNA using a miniPCR and a simple plate reader. In the current context of the COVID-19 pandemics, the importance of communicating this result does not reside in its novelty but in its practicality. In our experiments, we have used the three sets of primers designed and recommended by the CDC to identify the presence of SARS-CoV-2, the causal agent of COVID-19. These primer pairs, aimed at identifying three different regions encoding for the N protein of SARS-CoV-2, have been widely validated and used for diagnostic purposes in actual COVID-19 patients, Here we simply translated widely tested protocols

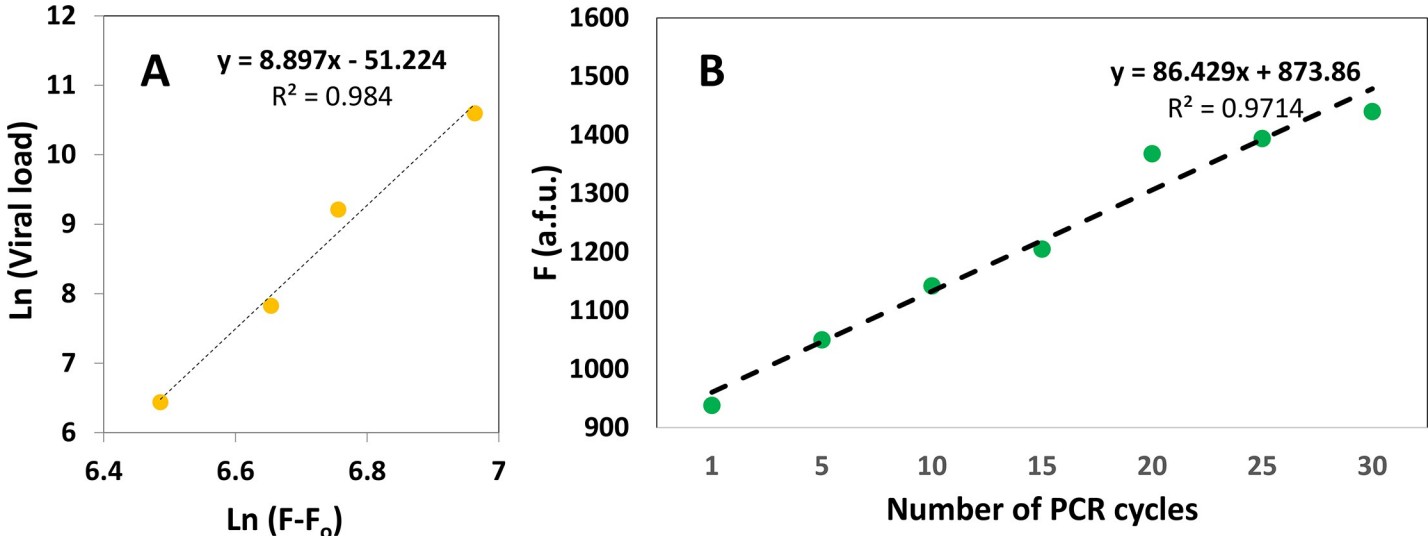

**Fig 3. Potential use of a plate reader for quantitation of the initial viral load in a sample and the extent of amplification.** (A) Linear relationship between the natural logarithm of the fluorescence reading and the natural logarithm of the viral load. (B) Results of the simulation of real time PCR in a microplate reader.

from the framework of an RT-qPCR apparatus (the gold standard platform recommended for analyzing and confirming positive cases) to execution in a miniaturized and already commercial POC thermal cycler. While the cost of a commercial RT-qPCR apparatus falls in the range of $10,000 to $40,000 USD, the commercial value of the miniPCR is under $800 USD. This difference is significant, especially when considering the need for rational investment of resources during an epidemic crisis.

While the quantitative capabilities of testing in a RT-QPCR platform are undisputable, the capacity of many countries for rapid, effective, and massive establishment of diagnostic centers based on RT-qPCR is questionable. The current pandemic scenarios experienced in the USA, Italy, France, and Spain, among others, have crudely demonstrated that centralized labs are not an ideal solution during emergencies. Portable diagnostic systems may provide the required flexibility and speed of response that RT-qPCR platforms cannot deliver.

To further illustrate the deterministic and quantitative dependence between the concentration of amplification product and the fluorescence signal, as measured in a plate reader, we simulated some real-time amplification experiments. To that end, we conducted amplification reactions using initial amounts of $4 \times 10^4$ copies of synthetic SARS-CoV-2 in the miniPCR cycler. We added the intercalating agent, EvaGreen Dye, to the reaction mix at the initial time and extracted samples after 1, 5, 10, 15, 20, 25, and 30 PCR cycles. The fluorescence from these samples was then measured in a plate reader. We observed a linear increase in fluorescence as more PCR cycles were performed (Fig 3B); this highlights the quantitative nature of the intercalating reaction.

Our results suggest that using a commercial plate reader to determine the extent of advance of PCR amplifications is a practical, reliable, reproducible, and robust alternative to the use of gel electrophoresis. Moreover, fluorescence reading of PCR products may lead to precise quantification of viral loads.

## Conclusions

The current COVID-19 pandemic has crudely demonstrated that our available methods of detection have severe limitations in terms of cost-efficiency, scalability, and amenability for rapid implementation. Developing and well-developed countries have experienced severe difficulties in intensifying diagnostics, a required condition to stop the pandemic advance in densely populated cities. Since time is the most limiting factor in emergencies, the integration of well-developed and commercially available technologies becomes an obvious, expedient, and cost-effective first line of defense during epidemic events. Our research extends the validation of the miniPCR technology to the as-yet-unexplored topic of detection of COVID-19. Furthermore, we suggest the combined use of the miniPCR and a conventional well-plate reader as a reliable strategy that can expand the testing capabilities of RT-qPCR.

We used the set of primers developed by the CDC and recommended by the WHO for conducting the standard PCR diagnostics of COVID-19. These primers target three different regions of the viral nucleic acids encoding for the N protein. In our experiments, we corroborate that the miniPCR apparatus is capable of amplifying small amounts of SARS-CoV-2 synthetic nucleic acids. We were able to detect and amplify 64 copies of genes encoding for the N protein of SARS-CoV-2. In the context of the COVID-19 pandemics, the use of the miniPCR thermocycler may be a valuable tool to intensify diagnostics by providing relevant advantages of higher portability, lower capital cost, and easier operation than can be achieved with other RT-qPCR platforms. We found the miniPCR® to be simple and intuitive to use; these are important attributes that would facilitate the widespread adoption of any diagnostic technology.

Moreover, the combined use of the miniPCR thermocycler and a 96-well plate reader enables the possibility of obtaining immediate readings of the amplification products, thereby providing faster (and potentially quantitative) diagnostic results in shorter times than when gel electrophoresis techniques are used. Therefore, the integration of these two already commercially available devices—a miniPCR thermocycler and a 96-well plate reader—has great potential for use during epidemic emergencies.

## Supporting information

**S1 File.**
(XLSX)

**S1 Fig. Uncropped and unadjusted images of gels presented in Fig 2A: Images of agarose gel electrophoresis of the DNA amplification product generated by targeting a region of the sequence coding for SARS-CoV-2 N protein.** PCR was performed using a miniPCR® thermocycler. The primer set N1 was used. The initial template amount was gradually increased from left to right: negative control (lane 1), 625 copies (lane 2), $2.5 \times 10^3$ (lane 3), $1.0 \times 10^4$ (lane 4), repetition of $1.0 \times 10^4$ (lane 5), $4.0 \times 10^4$ (lane 6), $2.0 \times 10^5$ DNA copies (lane 7), and molecular weight ladder (lane 8).
(TIFF)

**S2 Fig. Uncropped and unadjusted images of gels presented in Fig 2B: Images of agarose gel electrophoresis of the DNA amplification product generated by targeting a region of the sequence coding for SARS-CoV-2 N protein.** PCR was performed using a miniPCR® thermocycler. The primer set N2 was used. The initial template amount was gradually increased from left to right: negative control (lane 1), 625 copies (lane 2), $2.5 \times 10^3$ (lane 3), $1.0 \times 10^4$ (lane 4), repetition of $1.0 \times 10^4$ (lane 5), $4.0 \times 10^4$ (lane 6), $2.0 \times 10^5$ DNA copies (lane 7), and molecular weight ladder (lane 8).
(TIFF)

**S3 Fig. Uncropped and unadjusted images of gels presented in Fig 2C: Images of agarose gel electrophoresis of the DNA amplification product generated by targeting a region of the sequence coding for SARS-CoV-2 N protein.** PCR was performed using a miniPCR® thermocycler. The primer set N3 was used. The initial template amount was gradually increased from left to right: negative control (lane 1), 625 copies (lane 2), $2.5 \times 10^3$ (lane 3), $1.0 \times 10^4$ (lane 4), repetition of $1.0 \times 10^4$ (lane 5), $4.0 \times 10^4$ (lane 6), $2.0 \times 10^5$ DNA copies (lane 7), and molecular weight ladder (lane 8).
(TIFF)

## Author Contributions

**Conceptualization:** Mario Moisés Alvarez.

**Data curation:** Itzel Montserrat Lara-Mayorga, Sergio Omar Martínez-Chapa.

**Formal analysis:** Everardo González-González, Grissel Trujillo-de Santiago, Itzel Montserrat Lara-Mayorga, Mario Moisés Alvarez.

**Funding acquisition:** Grissel Trujillo-de Santiago, Mario Moisés Alvarez.

**Investigation:** Everardo González-González, Itzel Montserrat Lara-Mayorga.

**Methodology:** Everardo González-González, Itzel Montserrat Lara-Mayorga, Mario Moisés Alvarez.

**Resources:** Sergio Omar Martínez-Chapa, Mario Moisés Alvarez.

**Supervision:** Grissel Trujillo-de Santiago, Itzel Montserrat Lara-Mayorga, Sergio Omar Martínez-Chapa, Mario Moisés Alvarez.

**Validation:** Mario Moisés Alvarez.

**Writing – original draft:** Everardo González-González, Mario Moisés Alvarez.

**Writing – review & editing:** Grissel Trujillo-de Santiago, Itzel Montserrat Lara-Mayorga, Sergio Omar Martínez-Chapa, Mario Moisés Alvarez.

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
