## [Decision Letter · Decision Letter 0]

21 May 2020

PONE-D-20-10541

Portable and accurate diagnostics for COVID-19: Combined use of the miniPCR® thermocycler and a well-plate reader for SARS-CoV-2 virus detection

PLOS ONE

Dear Dr. Mario Moisés Alvarez, 

Thank you for submitting your manuscript to PLOS ONE. After careful consideration, we feel that it has merit but does not fully meet PLOS ONE’s publication criteria as it currently stands. Therefore, we invite you to submit a revised version of the manuscript that addresses the points raised during the review process.

ACADEMIC EDITOR:  

This manuscript describes a diagnostic method for COVID-19. It could be useful in developing countries. However, the authors should to address the issue of reduced specificity in the Discussion section as a limitation of the diagnostic system.

We would appreciate receiving your revised manuscript by Jul 05 2020 11:59PM. To enhance the reproducibility of your results, we recommend that if applicable you deposit your laboratory protocols in protocols.io, where a protocol can be assigned its own identifier (DOI) such that it can be cited independently in the future. For instructions see: http://journals.plos.org/plosone/s/submission-guidelines#loc-laboratory-protocols

We look forward to receiving your revised manuscript.

Kind regards,

Shawky M. Aboelhadid, PhD

Academic Editor

PLOS ONE

Journal Requirements:

2. We noticed minor instances of text overlap with the following previous publication, by some of your co-authors, which need to be addressed:

(1) https://journals.plos.org/plosone/article?id=10.1371%2Fjournal.pone.0215642

The text that needs to be addressed involves the Introduction and Discussion section.

In your revision please ensure you cite all your sources (including your own works), and quote or rephrase any duplicated text outside the methods section. Further consideration is dependent on these concerns being addressed."

3. To comply with PLOS ONE submission guidelines, in your Methods section, please provide additional information regarding your statistical analyses. For more information on PLOS ONE's expectations for statistical reporting, please see https://journals.plos.org/plosone/s/submission-guidelines.#loc-statistical-reporting.

Reviewers' comments:

Reviewer's Responses to Questions

**Comments to the Author**

1. Is the manuscript technically sound, and do the data support the conclusions?

Reviewer #1: Partly

Reviewer #2: No

Reviewer #3: Yes

2. Has the statistical analysis been performed appropriately and rigorously? 

Reviewer #1: Yes

Reviewer #2: Yes

Reviewer #3: Yes

3. Have the authors made all data underlying the findings in their manuscript fully available?

Reviewer #1: Yes

Reviewer #2: Yes

Reviewer #3: Yes

4. Is the manuscript presented in an intelligible fashion and written in standard English?

Reviewer #1: No

Reviewer #2: Yes

Reviewer #3: Yes

5. Review Comments to the Author

Reviewer #1: Comment 1:

Please write abbreviations in full-from when they first appear in the text (Eg: Line 51: COVID-19 should be defined as coronavirus 2019; Line 52: SARS-CoV-2 should be defined as severe acute respiratory syndrome coronavirus 2; etc.).

Comment 2:

As you mentioned (Line 62-65: For example, for the last two pandemic events involving influenza A/H1N1/2009 and COVID-19 [20], the Centers for Disease Control (CDC) and the World Health Organization (WHO) recommended RT-qPCR methods as the gold standard for official detection of positive cases) RT-qPCR is the gold method for the detection of virus and real-time PCR system with approximate size of 100 mm × 60 mm × 33 mm (smaller than the apparatus described in your article) has been described (https://dx.doi.org/10.1039%2Fc5lc01415h). Therefore, can you please specify the advantages of using the apparatus you describe as compared to this one?

Comment 3:

Although the PCR machine you describe is compact, additional machinery (power supply, gel tank and imaging system for first method; microplate reader for third method) is required for the confirmation of the presence of the virus. Considering that the additional equipment is required, would it not defeat the purpose of using a compact portable PCR machine?

Comment 4:

In the paper at line 188-190, you stated “The miniPCR® was able to generate a visible band of amplification products for all three primer sets and across the whole range of synthetic viral loads”, but lane 3 (625 copies) in figure 2A-C is not visible. Please clarify.

Comment 5:

Line 182-184: “actual viral load of COVID-19 in nasal swabs from patients has been estimated to fall within the range of 105 to 106 viral copies per mL”. Line 113: 1μL of DNA template was used. Therefore, the estimated DNA that will be used from the nasal swab is (1μL from 105 to 106/mL) 100-1000 copies. Is your proposed method reliable to detect the DNA in this range?

Comment 6:

Line 207-209: “Figure 2a shows the fluorescence readings associated with the analysis of the different dilutions of synthetic SARS-CoV-2 samples previously revealed by gel electrophoresis”. However, figure 2a is a gel image. Please rectify.

Comment 7:

Figure 3 is not mentioned in-text. Please rectify.

Reviewer #2: The study entitled” Portable and accurate diagnostics for COVID-19: Combined use of the miniPCR® thermocycler and a well-plate reader for SARS-CoV-2 virus detection” described the combined use of the miniPCR® and a commercial well-plate reader as a diagnostic system for SARS-CoV-2. They amplified three amplicons with the N gene of the SARS-CoV-2 and added

EvaGreen® Dye on DNA samples prior to amplification. The extent of positivity was measured after amplification using ELISA reader. Although considered a none sense modification in the developed countries where TaqMan real time qPCR is considered the gold standard protocol, it may be useful only in the developing countries to replace the SYBR green based realtime qPCR but it produces less specific results and one cannot depend on it on the virus diagnosis. It is known that the diagnostic RT-qPCR is mainly based on TaqMan protocol that confers more specificity than SYBR green based assay.

Reviewer #3: This paper describes a portable PCR machine but additional materials (power supply, gel tank and imaging system or microplate reader) is required. The MS is well written and describes an important study.

Few comments: the authors said that a visible band was seen for all concentrations while a band cannot be seen in the image for 625 copies

Line 207-209: “Figure 2a shows the fluorescence readings associated with the analysis of the different dilutions of synthetic SARS-CoV-2 samples previously revealed by gel electrophoresis”. However, figure 2a is a gel image.

Figure 3 is not mentioned in the ms.

6. PLOS authors have the option to publish the peer review history of their article (what does this mean?). If published, this will include your full peer review and any attached files.

Reviewer #1: No

Reviewer #2: No

Reviewer #3: No

---

## [Author Response · Author response to Decision Letter 0]

5 Jul 2020

Response to Reviewers

Reviewer #1: 

Comment 1:

Please write abbreviations in full-from when they first appear in the text (Eg: Line 51: COVID-19 should be defined as coronavirus 2019; Line 52: SARS-CoV-2 should be defined as severe acute respiratory syndrome coronavirus 2; etc.).

We thank you for your accurate observation. In this revised version we have written in full all abbreviations the first time they are used. For example:

In line 54 we state: “Nevertheless, the coronavirus disease 2019 (COVID-19) pandemic caused by the severe acute respiratory syndrome coronavirus 2 (SARS-CoV-2) [9] has broadsided most well developed and developing countries…”

In line 66 we wrote: “the Centers for Disease Control (CDC) and the World Health Organization (WHO) recommended the reverse transcription quantitative polymerase chain reaction (RT-qPCR) as the gold standard for official detection of positive cases.”

Comment 2:

As you mentioned (Line 62-65: For example, for the last two pandemic events involving influenza A/H1N1/2009 and COVID-19 [20], the Centers for Disease Control (CDC) and the World Health Organization (WHO) recommended RT-qPCR methods as the gold standard for official detection of positive cases) RT-qPCR is the gold method for the detection of virus and real-time PCR system with approximate size of 100 mm × 60 mm × 33 mm (smaller than the apparatus described in your article) has been described (https://dx.doi.org/10.1039%2Fc5lc01415h). Therefore, can you please specify the advantages of using the apparatus you describe as compared to this one?

The reviewer is absolutely right. There are reports of PCR and RT-qPCR systems smaller than the miniPCR and their functioning has been demonstrated as prototypes. However, these systems are not commercially available. In the context of an emergency such as the current COVID-19 pandemic, we believe that the first line of defense in diagnostics has to be based on fully developed (preferably commercially available) equipment. Among the various compact PCR systems currently known, the miniPCR offers precisely the highly relevant advantage of being a fully tested commercial product. To address your comment, in the revised version of the manuscript, we now refer to the development that you have mentioned, among several other POC RT-qPCR versions:

Line 72: “Several different versions of compact PCR platforms (some of them q-PCR systems) have been described recently in the scientific literature [24-27]. Unfortunately, most of these devices have not yet reached the market. During epidemic emergencies, resourcing of incompletely developed technologies is impractical, and the use of commercially available diagnostic platforms becomes the first and arguably the most cost-efficient line of defense.”

Comment 3:

Although the PCR machine you describe is compact, additional machinery (power supply, gel tank and imaging system for first method; microplate reader for third method) is required for the confirmation of the presence of the virus. Considering that the additional equipment is required, would it not defeat the purpose of using a compact portable PCR machine?

Your point is well taken. As stated in the title and abstract of the original version of the manuscript, the strategy that we propose and demonstrate here is the combined used of the miniPCR and a conventional well-plate reader. 

Line 27: “Here, we demonstrate the use of the miniPCR®, a commercial compact and portable PCR device recently available on the market, in combination with a commercial well-plate reader as a diagnostic system for detecting genetic material of the severe acute respiratory syndrome coronavirus 2 (SARS-CoV-2), the causal agent of COVID-19.”

The blueGel� and an alternative electrophoretic unit referred to in our manuscript were used only to demonstrate that the amplification products are properly generated in the miniPCR unit. In an actual scenario of use, only the miniPCR and the plate reader will be needed, instead of a RT-qPCR unit. 

In this revised version of the manuscript, we also explain this:

In line 321: “Moreover, the combined use of the miniPCR thermocycler and a 96-well plate reader enables the possibility of obtaining immediate readings of the amplification products, thereby providing faster (and potentially quantitative) diagnostic results in shorter times than when gel electrophoresis techniques are used. Therefore, the integration of these two already commercially available devices—a miniPCR thermocycler and a 96-well plate reader—has great potential for use during epidemic emergencies.”

Moreover, the current situation that we are experiencing in different parts of the world is vivid evidence of the need for more portable, but also more flexible and cost-effective, PCR systems. As we explain in this manuscript, the combination of miniPCR and a reader plate is convenient due to their superior portability, but also to their flexibility, cost, and ease of use. 

In attention to your comment, in this revised version of the manuscript, we wrote:

Line 315: “In the context of the COVID-19 pandemics, the use of the miniPCR thermocycler may be a valuable tool to intensify diagnostics by providing relevant advantages of higher portability, lower capital cost, and easier operation than can be achieved with other RT-qPCR platforms. We found the miniPCR® to be simple and intuitive to use; these are important attributes that would facilitate the widespread adoption of any diagnostic technology.”

The proposed strategy of combining miniPCR and a plate reader, as recently referred in a blog at nature.com (please see below), is related to the concept of simplicity that leads to effective solutions: “Think Simple”:

https://bioengineeringcommunity.nature.com/posts/think-simple

This blog mentions that our contribution aims to simplify and scale up diagnostic capacities. We will most probably acquire a miniPCR and a plate reader to equip a conventional biolab for testing rather than a quantitative PCR machine. 

Comment 4:

In the paper at line 188-190, you stated “The miniPCR® was able to generate a visible band of amplification products for all three primer sets and across the whole range of synthetic viral loads”, but lane 3 (625 copies) in figure 2A-C is not visible. Please clarify.

We apologize. In this revised version of the manuscript, we show (Figure 2A-C) gels in which the product of amplification corresponding to 625 copies can be clearly observed. 

Comment 5:

Line 182-184: “actual viral load of COVID-19 in nasal swabs from patients has been estimated to fall within the range of 105 to 106 viral copies per mL”. Line 113: 1μL of DNA template was used. Therefore, the estimated DNA that will be used from the nasal swab is (1μL from 105 to 106/mL) 100-1000 copies. Is your proposed method reliable to detect the DNA in this range?

Yes, our technique is a compact embodiment of conventional PCR that preserves the sensitivity of PCR. In this revised version, we were able to detect low copy numbers. We demonstrated this using gel electrophoresis experiments (Figure 2A-C) and results obtained using a plate reader (Figure 2D-F). 

We believe that an additional note is pertinent. Indeed, a patient may have between 105 and 106 copies of viral RNA per mL of nasal swab sample. This would mean that RT-qPCR would be able to find per 105–106 copies of viral RNA in a volume of RNA extract (typically a few microliters) derived from 1 mL of actual nasal sample. Indeed, readings of 104 to 105 copies of SARS-CoV-2 RNA in the final volume placed in the thermocycler (typically 200–250 µL) are not uncommon in RT-qPCR testing.

Comment 6:

Line 207-209: “Figure 2a shows the fluorescence readings associated with the analysis of the different dilutions of synthetic SARS-CoV-2 samples previously revealed by gel electrophoresis”. However, figure 2a is a gel image. Please rectify.

We apologize. We have corrected this mistake. In this revised version have corrected this:

In line 205: “Figures 2D-F shows the fluorescence readings associated with the analysis of the different dilutions of synthetic SARS-CoV-2 samples previously revealed by gel electrophoresis.”

Comment 7:

Figure 3 is not mentioned in-text. Please rectify.

We apologize for this mistake; in this revised version we appropriately mentioned Figure 3.

In line 252:“where Fo is the fluorescence reading exhibited by a blank (i.e., a negative sample prepared and processed in the same way than the positive samples) and α = 8.897 (as determined by fitting of the data presented in Figure 3A).”

In line 290 we state: “We observed a linear increase in fluorescence as more PCR cycles were performed (Figure 3B); this highlights the quantitative nature of the intercalating reaction.”

Reviewer #2: 

Comment 1:

The study entitled” Portable and accurate diagnostics for COVID-19: Combined use of the miniPCR® thermocycler and a well-plate reader for SARS-CoV-2 virus detection” described the combined use of the miniPCR® and a commercial well-plate reader as a diagnostic system for SARS-CoV-2. They amplified three amplicons with the N gene of the SARS-CoV-2 and added

EvaGreen® Dye on DNA samples prior to amplification. The extent of positivity was measured after amplification using ELISA reader. Although considered a none sense modification in the developed countries where TaqMan real time qPCR is considered the gold standard protocol, it may be useful only in the developing countries to replace the SYBR green based realtime qPCR but it produces less specific results and one cannot depend on it on the virus diagnosis. It is known that the diagnostic RT-qPCR is mainly based on TaqMan protocol that confers more specificity than SYBR green based assay.

Your point is well taken. We are not suggesting replacing standard RT-qPCR. We are suggesting an addition to the portfolio of options that we have now available to detect SARS-CoV-2. Six months after the onset of the COVID-19 pandemics, we have seen, in developing (México, Brazil, Chile, India) as well developed (USA, England, Italy, Spain) countries, that conventional RT-qPCR has been vastly insufficient. The problem is not accuracy or reliability, where RT-qPCR would be unsurpassed for sure; the problem is flexibility, portability, ease of use, and, ultimately, the scalability of the testing effort. The axis of our proposal is the combined use of two low-cost and simple pieces of equipment to conduct quantitative PCR, thereby enhancing the capacity of testing that has been insufficient worldwide so far. 

In attention to your comment, in this revised version (line 299) we explain:

“The current COVID-19 pandemic has crudely demonstrated that our available methods of detection have severe limitations in terms of cost-efficiency, scalability, and amenability for rapid implementation. Developing and well-developed countries have experienced severe difficulties in intensifying diagnostics, a required condition to stop the pandemic advance in densely populated cities. Since time is the most limiting factor in emergencies, the integration of well-developed and commercially available technologies becomes an obvious, expedient, and cost-effective first line of defense during epidemic events. Our research extends the validation of the miniPCR technology to the as-yet-unexplored topic of detection of COVID-19. Furthermore, we suggest the combined use of the miniPCR and a conventional well-plate reader as a reliable strategy that can expand the testing capabilities of RT-qPCR.”

Reviewer #3: 

This paper describes a portable PCR machine but additional materials (power supply, gel tank and imaging system or microplate reader) is required. The MS is well written and describes an important study.

We deeply thank you for your comment on the value of our contribution. 

Few comments: the authors said that a visible band was seen for all concentrations while a band cannot be seen in the image for 625 copies

We apologize. In this revised version of the manuscript, we have replaced the gel images that were unsatisfactory. The new gels more clearly show the amplification products corresponding to 625 copies of viral RNA (Figure 2A-C). 

Line 207-209: “Figure 2a shows the fluorescence readings associated with the analysis of the different dilutions of synthetic SARS-CoV-2 samples previously revealed by gel electrophoresis”. However, figure 2a is a gel image.

We apologize; we have gladly corrected this mistake and in this revised version (line 205) we state:

“Figures 2D-F shows the fluorescence readings associated with the analysis of the different dilutions of synthetic SARS-CoV-2 samples previously revealed by gel electrophoresis.”

Figure 3 is not mentioned in the ms.

We apologize for this mistake; in this revised version, we appropriately mentioned Figure 3.

In line 252: “where Fo is the fluorescence reading exhibited by a blank (i.e., a negative sample prepared and processed in the same way than the positive samples) and α = 8.897 (as determined by fitting of the data presented in Figure 3A).”

In line 290: “We observed a linear increase in fluorescence as more PCR cycles were performed (Figure 3B); this highlights the quantitative nature of the intercalating reaction.”

---

## [Decision Letter · Decision Letter 1]

28 Jul 2020

Portable and accurate diagnostics for COVID-19: Combined use of the miniPCR thermocycler and a well-plate reader for SARS-CoV-2 virus detection

PONE-D-20-10541R1

Dear Dr. Mario Moisés Alvarez,

We’re pleased to inform you that your manuscript has been judged scientifically suitable for publication and will be formally accepted for publication once it meets all outstanding technical requirements.

Kind regards,

Shawky M. Aboelhadid, PhD

Academic Editor

PLOS ONE

Additional Editor Comments (optional):

Reviewers' comments:

Reviewer's Responses to Questions

**Comments to the Author**

1. If the authors have adequately addressed your comments raised in a previous round of review and you feel that this manuscript is now acceptable for publication, you may indicate that here to bypass the “Comments to the Author” section, enter your conflict of interest statement in the “Confidential to Editor” section, and submit your "Accept" recommendation.

Reviewer #3: All comments have been addressed

2. Is the manuscript technically sound, and do the data support the conclusions?

Reviewer #3: Yes

3. Has the statistical analysis been performed appropriately and rigorously? 

Reviewer #3: Yes

4. Have the authors made all data underlying the findings in their manuscript fully available?

Reviewer #3: Yes

5. Is the manuscript presented in an intelligible fashion and written in standard English?

Reviewer #3: Yes

6. Review Comments to the Author

Reviewer #3: All comments have been addressed. You may like to update the data regarding the number of cases in the intro/abstract.

7. PLOS authors have the option to publish the peer review history of their article (what does this mean?). If published, this will include your full peer review and any attached files.

Reviewer #3: No

---

## [Editor Report · Acceptance letter]

3 Aug 2020

PONE-D-20-10541R1 

Portable and accurate diagnostics for COVID-19: Combined use of the miniPCR thermocycler and a well-plate reader for SARS-CoV-2 virus detection 

Dear Dr. Alvarez:

I'm pleased to inform you that your manuscript has been deemed suitable for publication in PLOS ONE. Congratulations! Your manuscript is now with our production department. 

Kind regards, 

on behalf of

Professor Shawky M. Aboelhadid 

Academic Editor

PLOS ONE